# Reciprocal Regulation of Hippo and WBP2 Signalling—Implications in Cancer Therapy

**DOI:** 10.3390/cells10113130

**Published:** 2021-11-11

**Authors:** Yvonne Xinyi Lim, Hexian Lin, Sock Hong Seah, Yoon Pin Lim

**Affiliations:** 1Integrative Sciences and Engineering Programme, National University of Singapore, Singapore 119077, Singapore; yvonnelimxy@u.nus.edu (Y.X.L.); hexian_lin@u.nus.edu (H.L.); seahsh@u.nus.edu (S.H.S.); 2Department of Biochemistry, National University of Singapore, Singapore 117596, Singapore; 3Mechanobiology Institute, National University of Singapore, Singapore 117411, Singapore

**Keywords:** hippo pathway, cancer, oncogene, therapeutics, reciprocal regulation, WW-domain binding protein 2

## Abstract

Cancer is a global health problem. The delineation of molecular mechanisms pertinent to cancer initiation and development has spurred cancer therapy in the form of precision medicine. The Hippo signalling pathway is a tumour suppressor pathway implicated in a multitude of cancers. Elucidation of the Hippo pathway has revealed an increasing number of regulators that are implicated, some being potential therapeutic targets for cancer interventions. WW domain-binding protein 2 (WBP2) is an oncogenic transcriptional co-factor that interacts, amongst others, with two other transcriptional co-activators, YAP and TAZ, in the Hippo pathway. WBP2 was recently discovered to modulate the upstream Hippo signalling components by associating with LATS2 and WWC3. Exacerbating the complexity of the WBP2/Hippo network, WBP2 itself is reciprocally regulated by Hippo-mediated microRNA biogenesis, contributing to a positive feedback loop that further drives carcinogenesis. Here, we summarise the biological mechanisms of WBP2/Hippo reciprocal regulation and propose therapeutic strategies to overcome Hippo defects in cancers through targeting WBP2.

## 1. The Hippo Pathway

### 1.1. Overview of the Hippo Pathway

The Hippo pathway was first discovered in *Drosophila melanogaster* and later found to be highly conserved across several higher-order vertebrates including *Homo sapiens*. It consists of an extensive cascade of regulator proteins that culminate signals to inhibit the activity of the downstream yes-associated protein (YAP) and its paralog transcriptional coactivator with the PDZ-binding motif (TAZ) (also known as WWTR1). Both YAP and TAZ are transcriptional co-regulators that do not contain DNA-binding domains. When localised in the nucleus, YAP and TAZ cooperate with other transcriptional factors (TFs), the most prominent being the TEAD family [1,2], to direct cellular processes such as proliferation, apoptosis, cell cycle progression and organ size control [3,4,5,6,7,8,9,10]. Other TFs that have been found to interact with YAP/TAZ include the SMAD and RUNX (Runt-domain transcription factors) TF families [11,12,13,14].

The activity of YAP/TAZ is predominantly regulated by the core Hippo signalling cassette, which is comprised of the mammalian STE20-like protein kinase 1/2 (MST1/2; also known as STK4/3, respectively), the large tumour suppressor 1/2 (LATS1/2) and adaptor proteins Salvador homologue 1 (SAV1) and MOB kinase activator 1A/B (MOB1A/B). Activation of MST1/2 results in LATS1/2 phosphorylation with the aid of the adaptor proteins SAV1 and MOB [15,16]. Subsequently, activated LATS1/2 kinases phosphorylate YAP/TAZ (Figure 1). Upon phosphorylation, YAP/TAZ are inactivated due to cytoplasmic retention by binding with the 14-3-3 protein [3,17]. Additional phosphorylation of YAP/TAZ results in their proteasomal degradation that is mediated by the E3 ubiquitin ligase, beta transducing repeat-containing protein (βTrCP) [18,19]. On the other hand, emerging evidence has highlighted several neoteric modulators that can influence YAP/TAZ activity either directly or indirectly (Figure 1). For example, MAP4Ks, MEKK2/3 and TAOKs have been reported to phosphorylate LATS1/2 directly at their hydrophobic motifs, resulting in LATS1/2 activation in parallel to MST1/2 [20,21,22,23]. Furthermore, TAOKs can also phosphorylate and activate MST1/2 [24]. Other molecules, such as neurofibromin 2 (NF2), angiomotin (AMOT) and WWC and C2 domain-containing (WWC) proteins may modulate the Hippo pathway at multiple levels of regulation [23,25,26,27,28]. Intense elucidation of the diverse cellular mechanisms regulating YAP/TAZ phosphorylation, localization and transcriptional activities has led to a growing number of novel Hippo pathway regulators ranging from cytoskeletal, scaffolding and membrane receptor proteins, henceforth drastically expanding the Hippo signalling landscape (Figure 1) [29].

### 1.2. Dysregulation of the Hippo Pathway in Cancers

The link between Hippo signalling and carcinogenesis was first proposed through initial findings that showed that mutations of the Hippo pathway components promoted tissue overgrowth in Drosophila models [30,31,32,33,34]. Consistently, dysregulation of Hippo signalling has been reported in several human tumours arising from the breast, prostate, gastric, liver, kidney and colorectal tissues [35]. A comprehensive meta-analysis of >9000 tumour samples across 33 cancer types in The Cancer Genome Atlas (TCGA) showed that the Hippo pathway is one of the most commonly genetically altered pathways in cancers, along with Myc, Akt, TGFβ, p53 and Wnt signalling pathways [36]. Notably, *YAP*/*WWTR1* are highly amplified in cervical, lung, oesophageal, bladder and head and neck cancers, supporting their tumorigenic roles in squamous cell carcinoma (Figure 2) [37]. On the other hand, deep deletion and promoter methylation in *LATS1/2,* as well as inactivating mutations in *NF2* are frequently reported in malignant mesotheliomas, gynaecologic and lung cancers, in congruence with their tumour-suppressive properties (Figure 2) [36,37,38,39,40,41,42]. Gene fusion is another molecular alteration found in cancers with dysregulated Hippo signalling. An outcome of structural chromosomal aberration, driver fusion genes may be generated in certain cancer types due to the exchange of hyperactivated regulatory regions between two independent genes. For instance, chromosomal translocation of *WWTR1* and *CAMTA1* (calmodulin-binding transcription activator) have been identified in more than 90% of patients with epithelioid hemangioendothelioma, a subtype of vascular cancers (Figure 2). The resulting *WWTR1*/*CAMTA1* fusion gene is controlled by the *WWTR1* gene promoter to encode a chimeric oncogenic transcription factor that constitutively activates TEAD target gene expression [43]. Additional gene fusions have also been identified in other vascular and lung cancers involving *YAP, TFE3* (transcription factor E3), *WWTR1, NF2, LATS1* and *LATS2* [44,45] (Figure 2). Together, these studies suggest that genetic aberration is an important contributing factor to unrestrained Hippo signalling in cancers and serves as a potential therapeutic target for tumours with Hippo pathway dysregulation.

In line with the mutational analysis, nuclear YAP/TAZ staining detected through immunohistochemistry negatively correlated with the overall survival and prognosis of patients with multiple tumour types [46]. Correspondingly, the YAP/TAZ target score, a representative measure of YAP/TAZ transcriptional activity, significantly predicted poorer overall survival in 8 out of 33 cancer types in the TCGA Pan-cancer Atlas [37]. On the other hand, downregulation of LATS1/2 correlated with a poorer prognosis (such as tumour, node and metastasis (TNM) stage and distal metastasis) in gastric cancer [46,47,48]. In colorectal cancer, the mRNA and protein levels of MST1 and LATS2 were reduced in tumour tissues compared to adjacent normal tissues and the converse trend was observed for YAP and TEAD mRNA and protein levels. Additionally, reduced mRNA expression of *MST1* and *LATS2*, followed by increased mRNA levels of *YAP*, *TAZ* and *TEAD1*, were observed in colorectal tumours of progressive TNM stages [49] (Figure 2). Functional studies in cancer cell lines and murine tumour models have demonstrated that the loss of the Hippo pathway components and the overexpression (or hyperactivation) of oncogenic YAP/TAZ closely correlates with the hallmarks of cancers; some of these hallmarks include hyperproliferation, metastasis, cancer stem cell maintenance, therapeutic resistance and even regulation of T cell-mediated anti-tumour responses [35,50,51].

Given the tumour-suppressive nature of Hippo signalling components and the oncogenic nature of its effector transcription co-regulators, YAP and TAZ, targeting this pathway is a compelling therapeutic strategy to subvert cancer initiation and progression [52,53]. Further delineation of the underlying mechanism regulating Hippo signalling is expected to provide new and druggable targets for exploitation in cancer therapies.

## 2. WW-Domain Binding Protein 2 (WBP2) and Its Functions in Cancers

Advents in molecular profiling platforms have provided an exhaustive list of putative oncogenes and tumour suppressors. The WW-domain binding protein 2 (WBP2) is an emerging oncoprotein that has been implicated in several cancers, especially in the breast [54]. Recently, WBP2 has also been associated with other diseases such as infertility and hearing loss [55,56]. In this section, we review the structure and function of WBP2, with a focus on its pro-tumorigenic roles in human cancers.

### 2.1. Domain and Structure of WBP2

The human WBP2 protein contains 261 amino acids, and as its name suggests, it recognises and binds to proteins containing WW domains. The WW domain is a semiconserved 38-amino acid module with two invariant tryptophan(W) residues that are 20 to 22 amino acids apart [57,58]. Found extensively in several structural and signal transduction proteins, the WW domain mediates protein–protein interaction through the recognition of proline-rich motifs and phosphorylated serine/threonine-proline sites. Structurally, WBP2 is characterised by two domains: the Glucosyltransferase Rab-like GTPase activator and the Myotubularins (GRAM) domain at the N terminus and a Proline-Rich region at the C terminus (Figure 3A). The GRAM domain is an intracellular domain that has functional implications in membrane-associated signal transduction [59]. On the other hand, the Proline-Rich region is comprised of three PPxY motifs that have been identified as important binding motifs for the WW domain [60]. These PPxY motifs are denoted as PY1, PY2 and PY3 and consist of the amino acid sequence of PPGY, PPPY and PPPY, respectively. Our laboratory has also identified two tyrosine phosphorylation sites (Y192 and Y231) located within the Proline-Rich region of WBP2 (Figure 3A). The phosphorylation of WBP2 can be induced by the epidermal growth factor receptor (EFGR) and Wnt signalling, and is a deterministic factor for WBP2 nuclear localization [61,62].

### 2.2. WBP2 Expression and Impact on Cancers

The oncogenic roles of WBP2 were initially studied in breast cancer and subsequently explored in other cancers such as squamous cell carcinoma, brain, liver, gastric and lung cancers. Figure 3B summarises some of the key evidence that positions WBP2 as a driver oncogene in various cancers.

The first evidence of WBP2 as a potential oncogene in breast cancer was observed by our laboratory. Using an isogenic MCF-10AT breast cancer progression model, WBP2 was shown to be hyperphosphorylated with increasing stage progression through phosphoproteomic analysis [63]. In vitro analysis in a breast cancer cell-line panel revealed an upregulation of the WBP2 protein level in the cancerous cell lines, as compared to the normal mammary cells. Interestingly, the same study also suggested that breast cancer cell lines from the more aggressive triple-negative breast cancer (TNBC) and human epidermal growth factor receptor 2-positive (HER2+) subtypes exhibit a higher expression of WBP2 relative to those from the less aggressive oestrogen receptor-positive (ER+) subtype [62]. Moreover, the clinical relevance of WBP2 was established through our immunohistochemistry (IHC) study of >400 breast tumour samples. WBP2 overexpression was detected in 85% of breast cancer samples compared to their normal mammary counterparts (Figure 3B). The overall and nuclear expression of WBP2 was also positively associated with tumour size and grade, while nuclear WBP2 levels were negatively correlated with overall and disease-free survival in breast cancer patients [62]. In another multicentre study involving 296 resected breast cancer tissues, tumours with higher WBP2 and HER2 expression were associated with worse overall and disease-free survival. This suggests that WBP2 is a prognostic marker in HER2+ breast cancer [64]. In line with these clinical studies, WBP2 has been reported to be frequently amplified or gained in multiple breast cancer databases including TCGA BRCA and METABRIC datasets [65]. Functionally, WBP2 has been found to promote several cancer phenotypes including growth [61,64,66], cell cycle progression [67], migration [61,68], invasion [61,62,68], epithelial mesenchymal transition [61] and chemotherapeutic resistance [64,67,69] in a number of breast cancer cell lines.

Besides breast cancer, WBP2 overexpression has also been observed in tumours of the liver, stomach, lung and skin, compared to adjacent normal tissues, through IHC experiments [70,71,72,73]. Furthermore, WBP2 status was positively associated with higher pathological stages and poorer survival in human glioma, gastric and non-small cell lung carcinoma (NSCLC) patients [25,71,72] (Figure 3B). In a similar fashion, elevated WBP2 protein expression was also observed in the more aggressive gastric cancer cell lines as compared to the less aggressive ones [71]. Concurrently, in vitro and in vivo studies demonstrated that WBP2 was a promoter of cancer proliferation, migration and invasion [70,71,72,73,74]. In glioma cells, WBP2 regulates α enolase-mediated glycolysis to aggravate cell proliferation and migration [74]. Collectively, these findings suggest that WBP2 is a bona fide oncogene in multiple cancer types besides breast cancer.

To obtain a more comprehensive view of WBP2 in cancer, our meta-analysis based on the TCGA Pan-Cancer Atlas revealed that WBP2 amplification was observed at high frequencies in approximately 78% of 32 cancer types (Figure 4) [68]. These include hepatocellular carcinoma, non-small cell lung cancer, glioma and gastric cancer [70,71,72], where WBP2 has already been reported to be functionally implicated. Moreover, WBP2 amplification was found in several other cancers, such as ovarian, cervical and pancreatic cancers. This highlights the immense potential of WBP2 to drive oncogenic processes in even more cancer types than what has been suggested in the current literature. Therefore, characterising the modes of action of WBP2 and its regulation in other solid tumours is a potential direction that can be undertaken in future research.

### 2.3. Molecular Roles of WBP2 in Oncogenic Signaling

Aberration of signal transduction in the cancer cells and their surrounding tumour microenvironment is key for driving cancer progression and metastasis. WBP2 has been projected to be a central node of the oncogenic signalling network through its multifarious actions. Primarily recognised as a transcriptional co-activator, alternative modes of action of WBP2, especially in non-transcriptional processes, are progressively unravelled. Here, we provide a brief overview of the transcriptional and non-transcriptional functions of WBP2 and assess how these molecular functions can mediate various oncogenic signalling pathways.

#### 2.3.1. Transcriptional Co-Activator Role

A staggering number of studies have substantiated the transcriptional co-activator role of WBP2 in a myriad of cancers. WBP2 has been shown to interact with several TFs such as YAP, TAZ, ER/PR and β catenin; this results in the augmentation of TF activities and their subsequent transcription of target genes [61,62,66,73,78,79,80].

The PY motifs located at the C terminus of WBP2 are key for the association of WBP2 with the WW domains of the abovementioned TFs (Figure 3A) [54]. Consequently, targeting WBP2 at its PY motif is a plausible therapeutic strategy to limit WBP2-driven cancer progression. As the majority of WBP2 resides in the cytoplasm, its nuclear entry is a pre-requisite to elicit its transcriptional co-regulatory activity. Studies have found that stimulation by extrinsic ligands such as Epidermal Growth Factor (EGF) and Wnt can induce WBP2 tyrosine phosphorylation, and such stimulation precedes its nuclear localization. In breast cancer, ectopic introduction of the phosphomimic mutant of WBP2 (Y192E, Y231E) was more efficacious in promoting WBP2 nuclear accumulation and the transcriptional activity of ER and β catenin [61].

Given that TFs are commonly altered in numerous cancers, targeting the protein–protein interaction between TFs and their associated transcriptional co-activators is a feasible approach to limit cancers [81]. Concurrently, there has been increased preclinical success in developing small-molecule inhibitors of protein–protein interactions [82]. Henceforth, the ability of WBP2 to mediate the activity of multiple TFs means that it has high potential to act as a therapeutic target in cancers with dysregulated TF activity.

A major caveat of targeting TF-co-factor interactions lies in the paucity of deep interaction pockets and hotspot residues that can serve as “druggable” targets for newly designed inhibitors. To overcome this challenge, alternative mechanisms such as the allosteric regulation of TF-co-factor interactions or the modulation of TF protein stability have been proposed [81]. Nonetheless, an in-depth biochemical understanding of WBP2-TFs interactions is crucial to advance this important research area into therapeutic application.

#### 2.3.2. Non-Transcriptional Co-Activator Role

WBP2 is undergoing a paradigm shift as recent studies have demonstrated its ramifications in relation to a diverse range of molecular actions, other than its transcriptional function. Firstly, WBP2 may act as an adaptor to directly or indirectly control enzymatic activities. In a study performed in glioma cells, WBP2 was found to induce the Akt pathway through its interaction with the α-enolase (ENO1) glycolytic enzyme [74]. In lung and gastric cancers, WBP2 has been shown to be a promoter of LATS1/2 kinase activities through interactions with critical regulatory components to inactivate Hippo signalling [71,72]. Secondly, WBP2 may be involved in microRNA (miRNA) pre-processing by disrupting the formation of the microprocessor complex (MPC) [83]. The miRNAs are small single-stranded non-coding RNAs that are known to play an important role in regulating gene expression. The miRNA biogenesis process is initiated by the transcription of DNA sequences into primary miRNAs (pri-miRNAs). Subsequently, the pri-miRNAs are trimmed into precursor miRNAs (pre-miRNAs) by the MPC complex and processed by Dicer. Dysregulated miRNA biogenesis is highly affiliated with cancer progression and the inhibition of miRNA biogenesis often results in a reduced expression of miRNAs that suppress oncogene levels in the cell [84,85]. Therefore, the involvement of WBP2 in MPC assembly provides another mode of action for WBP2 to drive carcinogenesis. Taken together, the discovery of the non-transcriptional roles of WBP2 provides new opportunities to target cancers with alterations in non-transcriptional processes. Given WBP2’s multimodal actions, we envisage that even more mechanistic actions of WBP2 will be revealed in a wider range of cancer types. This provides us with further insights to develop novel and promising targeting approaches to treat and manage multiple types of cancers.

## 3. Regulation of the Hippo Pathway by WBP2

The Hippo pathway has emerged as a signalling pathway that is regulated by WBP2 through its multimodal actions. Over the past 15 years, studies of the WBP2 oncoprotein have slowly shifted from the characterisation of its transcriptional co-activator role with YAP and TAZ to the elucidation of WBP2 as a significant regulator of the Hippo components (Figure 5). Here, we elaborate on the modes of action of WBP2 in regulating the Hippo signalling pathway in cancers.

### 3.1. WBP2 as a Transcriptional Co-Activator for YAP/TAZ

Analysis of the Hippo pathway regulators has highlighted a striking prevalence of proteins with WW domains [86,87,88]. It is therefore not surprising that WBP2 is highly implicated in the Hippo pathway through interactions with these WW domain-containing proteins. Interaction between WBP2 and YAP was first discovered using a functional screen of a cDNA expression library [60]. It was subsequently demonstrated that WBP2 acts as a co-factor of YAP through WW–PY domain interactions to drive YAP transcriptional activity, hence promoting YAP-induced proliferation in numerous cellular systems. In Drosophila melanogaster, it was demonstrated that WBP2 was essential for wing and eye tissue growth through its cooperation with Yorkie, the *Drosophila* homologue of YAP. Furthermore, WBP2 drives the overgrowth in Drosophila eye tissue in a manner dependent of warts (Lats1/2) deficiency [80]. In mammalian systems, the binding between WBP2 and YAP confers proliferative advantage in normal and malignant epidermal stem cells [73], as well as in breast cancer [79]. Similarly, WBP2 interacts with TAZ in a WW domain-dependent manner to drive TAZ-driven mammary cell transformation [66].

Contrary to its widely recognised oncogenic function, YAP may have a tumour-suppressive role through its association with the p73 transcription factor. YAP was found to bind to p73 and promote the latter’s transcriptional activity and the induction of apoptosis [89,90]. Furthermore, YAP is essential for maintaining p73 protein stability [91]. Interestingly, p73 was found to contain the PY motif which mediates its interaction with YAP in a similar fashion to WBP2 [91]. This suggests the possibility the WBP2 is a competitive inhibitor for YAP–p73 interaction. It will be intriguing to explore how WBP2 can interfere with the YAP–p73 interplay, and decipher the relevant translational implications in cancers.

### 3.2. WBP2 as an Adaptor for LATS1/2

In the context of lung and gastric cancers, WBP2 has been shown to negatively regulate LATS1 and LATS2 kinase activity through WW domain-dependent and WW domain-independent mechanisms, respectively [71,72]. WBP2 acts as a competitive inhibitor by binding to the WW domain of WWC3 protein in lung carcinoma cells to limit WWC3–LATS1 association [72]. Consistently, both LATS1 phosphorylation and TEAD transcriptional activity were reduced upon increased WBP2 expression [72].

On the other hand, WBP2 may bind directly to LATS2 to bring about inhibition of LATS2 phosphorylation and YAP/TEAD activity in gastric cancer cells, as reported by our laboratory [71]. LATS2 does not contain a WW domain, suggesting that the interaction between WBP2 and LATS2 is WW domain independent. Although we cannot rule out indirect binding, this study provides preliminary evidence that WBP2 may associate with another group of proteins in a manner that is independent of the presence of the WW domain. The authors demonstrated through in vitro pulldown assays that WBP2 likely associates with the kinase domain of LATS2 through its C terminus. However, the exact mode of action of WBP2 on regulating LATS2 activity remains unelucidated. A few hypotheses were formulated regarding the underlying mechanism: (1) WBP2 may allosterically inhibit LATS2 activity by stabilizing the inactive conformation of LATS2 and (2) WBP2 may act as an adaptor protein to connect LATS2 to an unknown regulator to contribute to LATS2 inactivation [71]. Further delineation of WBP2 and its implications in LATS1/2 kinase activity will provide more insights into the development of anti-WBP2 therapeutics for Hippo-dysregulated cancers.

Overall, this evidence suggests that the regulation of the Hippo pathway by the WBP2 oncoprotein is multilayered and multimodal; the modes of action of WBP2 on the Hippo pathway are summarised in Figure 6. However, although WBP2 was discovered to be a cognate ligand of YAP in 1995, the authors concede that the association of WBP2 with other components in the Hippo signalling pathway (i.e., LATS1/2 and MST1/2) has only been identified more recently (Figure 6). Notwithstanding that WBP2 has been implicated in the Hippo pathway in at least four different cancer types, including breast, gastric, skin and lung cancers, by independent groups [66,71,72,73,79], further studies of the interplay are required to clarify the role of WBP2 in the Hippo pathway.

## 4. Regulation of WBP2 by the Hippo Pathway

As a putative oncogene, the expression and activity of WBP2 is tightly controlled through a few distinct mechanisms [54]. The Hippo pathway is one of the signalling pathways that is involved in the regulation of WBP2. Here, we summarise the mechanistic insights gathered from current studies on the regulation of WBP2 expression by the Hippo pathway.

### 4.1. Post-Transcriptional Regulation of WBP2 by MST1/2 through miRNAs

Given that WBP2 is an integral factor for YAP/TAZ activation and the Hippo pathway inhibits YAP/TAZ, our laboratory hypothesised that the Hippo pathway can negatively regulate WBP2. Using TNBC cell lines, overexpression of MST1 and MST2 was observed to reduce WBP2 protein expression drastically in a LATS-independent manner [92]. Consistently, the higher expression of MST1/2 substantially attenuated WBP2-driven breast cancer growth in vivo and in vitro [92]. Activation of MST1/2 mediated the upregulation of Dicer, an enzyme required for pre-miRNA processing, and this resulted in the amelioration of WBP2 expression [92]. Therefore, this finding confirmed the existence of a MST1/2-Dicer pathway that influences WBP2 expression. To identify the miRNA that is directly involved in MST1/2-mediated WBP2 regulation, in silico analysis was performed. miR-23a was validated to be an authentic WBP2 regulator that is modulated by MST1/2-Dicer signalling. Our finding that WBP2 is regulated by miRNA is supported by the fact that a few miRNAs have been reported to negatively regulate WBP2 and mRNA levels in various cancer models, and these comprise of miR-613, miR-19a, miR-19b, miR-23a, miR-27a, miR-206, miR-485 and miR-613 [67,70,79,92]. A significant inverse association between WBP2 and MST1/2 or miR-23a was observed in clinical breast cancer specimens, hence substantiating the relevance of the MST1/2–Dicer–miR-23a–WBP2 axis in breast cancer [92].

On the other hand, WBP2 could impair the miRNA biogenesis process through the disruption of the MPC complex in the nucleus [83]. As a result, WBP2-targeting miRNAs were also inhibited [83]. Immunoprecipitation experiments in multiple breast cancer cell lines revealed the interaction of WBP2 with several MPC components, such as DGCR8, Drosha, DDX5 and DDX17. This led to the disassembly of MPC and a reduction in its miRNA processing capacity, as confirmed by biochemical MPC assays. Despite the fact that several MPC factors such as DGCR8 contain the WW domain, the PY motifs of WBP2 were not responsible for association with the MPC [83]. The exact mechanism underlying WBP2’s interaction with the MPC components remains to be investigated. In fact, the involvement of WBP2 in MPC assembly may provide mechanistic clues to explain the global suppression of miRNAs often observed in cancer patients [84,85]. In another study performed in our laboratory, WBP2 was reported to enhance the mRNA stability of BTRC in TNBC cells. Although not fully clarified, we conceive that this regulatory mechanism is likely to be through WBP2-mediated MPC activity. The induction of BTRC by WBP2 resulted in increased IκBα ubiquitin-mediated proteasomal degradation, and consequently NFκB activity, in turn driving TNBC migration and invasion [68].

Consistent with the actions of WBP2, the Hippo pathway has been implicated in the regulation of miRNA biogenesis. For example, nuclear YAP was observed to interact with DDX17, another critical component of the MPC complex, resulting in the sequestration of DDX17 from MPC and the subsequent suppression of pre-miRNA synthesis [93]. With substantial evidence supporting WBP2–YAP, WBP2–DDX17 and YAP–DDX17 interactions, it is conceivable that WBP2, YAP and DDX17 may associate with one another in a complex. On the other hand, nuclear YAP/TAZ accumulation mediated by low cell density is crucial for inducing Dicer stability and activity via the LIN28–Let-7 axis [94]. Whether WBP2 regulates Dicer through its interaction with YAP/TAZ is a question that has not yet been explored. It will be interesting to explore how the interplay between WBP2 and the Hippo pathway mediates miRNA biogenesis in cancers.

In summary, repression of Hippo signalling through MST1/2 is likely to augment WBP2 expression, generating a positive feedback loop to further suppress Hippo signalling and drive cancer progression (Figure 7). Retrospectively, WBP2 may induce its own expression indirectly through hindering miRNA biogenesis (Figure 7). Disruption of the miRNA processing process may lead to the attenuation of WBP2-specific miRNA and further escalate carcinogenesis processes mediated by WBP2.

### 4.2. Post-Translational Regulation of WBP2 through ITCH-Mediated Proteasomal Degradation

The discovery of ITCH (also referred to as atrophin-1 interacting protein 4 or AIP4) as the E3 ubiquitin ligase for LATS1 positioned ITCH as a regulator of Hippo signalling. ITCH appeared to exhibit oncogenic properties through mediating the degradation of the tumour suppressor LATS1 [95,96]. However, our laboratory found that ITCH can interact with the PPxY motif of WBP2 through its WW domain to induce the ubiquitin-mediated proteasomal degradation of WBP2. This suggests that ITCH may exert tumour-suppressive functions. Experimentally, we demonstrated that ITCH-mediated WBP2 degradation can be subverted by the overexpression of YAP and TAZ [62]. This is likely due to the competition between YAP/TAZ and ITCH to bind to the PPxY motifs of WBP2 [62]. Furthermore, oncogenic signals by Wnt3a and EGF shifted the equilibrium to favor oncogenic WBP2 expression by limiting ITCH–WBP2 interaction [62]. It is likely that ITCH can serve as a double-edged sword in different cellular contexts and more studies should be completed to better understand the interplay between ITCH, WBP2 and Hippo in a more physiological context, especially under ligand stimulation.

In a nutshell, while WBP2 is a potential regulator of the Hippo pathway, its expression may also be sophistically fine-tuned by members of the Hippo pathway (Figure 7A). Ultimately, the intricate loop involving WBP2 and the Hippo pathway provides multiple feedback mechanisms to boost the WBP2 level and inhibit Hippo signalling, resulting in a culminative equilibrium shift toward WBP2-driven cancer signalling (Figure 7B).

## 5. Exploring Other Potential Mechanisms of WBP2–Hippo Reciprocal Regulation

The accrued delineation of WBP2 and Hippo pathways has shed light on the reciprocal regulation of WBP2–Hippo signalling, and at the same time, provided substantial clues that may assist our future characterisation of the multifarious roles of WBP2 in Hippo signalling. Here, we highlight some potential interplays between WBP2 and the Hippo pathways through three putative modes of regulation: (1) E3 ubiquitin ligases, (2) mechanical proteins and (3) scaffold proteins.

### 5.1. Interplay between WBP2 and the Hippo Pathway through E3 Ubiquitin Ligases

Ubiquitination is a three-step enzymatic cascade reaction that facilitates the attachment of ubiquitin molecules to substrate proteins. The E3 ubiquitin ligases are a class of protein that is responsible for recognising the substrate proteins and mediating the transfer of ubiquitin molecules from E2 enzymes to the substrates. The HECT-type E3 ligase belongs to a group of E3 ligases that contain WW domains, a crucial motif to facilitate the recognition of its target substrates [97]. A few HECT-type E3 ligases have been implicated in Hippo signalling and these include ITCH, WWP1 and NEDD4 [62,98,99,100]; many of these ligases have been shown to bind to WBP2 [61,62,101,102,103]. While WBP2–ITCH interaction has been implicated in breast tumorigenesis [62], no study has yet explored the mechanistic and functional significance of WBP2–WWP1 and WBP2–NEDD4 interactions in cancers. Understanding the modes of action of WBP2 and its interaction with E3 ligases may provide novel avenues for cancer therapeutic strategies.

### 5.2. Interplay between WBP2 and the Hippo Pathway through Mechanical Proteins

Biological processes at the plasma membrane are increasingly implicated in Hippo signalling. Many upstream cues such as mechanical forces, cell adhesion, cell polarity and G protein-coupled receptors have been demonstrated to transmit their signals to Rho GTPase and actin cytoskeleton to regulate LATS1/2 and YAP/TAZ activities [104,105].

WBP2 has been reported to be implicated in cytoskeletal processes through omics and meta-analysis studies performed on numerous cellular contexts. Phosphoproteomics analysis across a gastric cell line panel revealed an upregulation of phosphorylated proteins involved in cytoskeleton organization along with WBP2 hyperphosphorylation and overexpression [71]. This suggests that WBP2 may be involved in cytoskeleton remodelling. In human epidermal cells, cell–cell contact inhibition was identified as the principal signal regulating WBP2–YAP interaction [73]. Therefore, mechanical forces may be a crucial mediator to link WBP2 with Hippo signalling.

Integrins are transmembrane proteins that sense signals from the ECM through multivariant interactions. These mechanical signals are then relayed to focal adhesion (FA), where they are integrated and further transmitted to a variety of signalling pathways, including the Hippo pathway [105]. Increased stiffness of the ECM can induce FAK and Src kinase activation via β1 integrins. Src kinase in turn inhibits LATS1/2 and activates YAP activity [106,107,108]. Alternatively, enhanced integrin–ECM interaction also prompted PAK1 phosphorylation via FAK, and activated PAK1 in turn inhibits NF2 and subsequent LATS1/2 activation [109]. On the other hand, the reduction in ECM stiffness is sensed by the transmembrane integrin complexes and relayed to GTPase RAP2, which interacts with MAP4K to activate LATS1/2 and suppress YAP [110]. YAP itself is a mechano-sensor that can transcriptionally activate genes related to integrins and FA complexes in response to changes in the ECM mechanics [111]. Collectively, the interplay between mechanical proteins and the Hippo pathway is well established and mediated via integrin signalling pathway. Therefore, an understanding of the crosstalk between WBP2 and integrin pathway may help us to decipher the interplay between WBP2 and Hippo signalling through mechanical proteins.

Several meta-analyses have illustrated the association between WBP2 and the integrin signalling pathway in the TCGA BRCA dataset [54,68]. Furthermore, a mass spectrometry analysis performed on glioma cells also identified VASP to be a putative binding partner of WBP2 [74]. VASP is an actin-binding protein involved in cytoskeletal organization and integrin-mediated Hippo signalling [112]. A recent study reported the importance of VASP in mediating the ECM–mediated β1-integrin-FAK signalling pathway to promote YAP/TAZ activation and liver metastasis in gastrointestinal (GI) cancer cells [112]. Therefore, it is possible that the cross-talk between integrin and Hippo signalling pathway can be regulated by WBP2–VASP interaction to drive tumorigenesis.

In a nutshell, the current evidence suggests that there may be an association between WBP2, Hippo pathways and mechanical proteins. However, this evidence remains primitive and warrants future investigation. We envision that WBP2 may act as a neoteric mechanotransduction modulator to regulate the Hippo signalling pathway in cancers.

### 5.3. Interplay between WBP2 and the Hippo Pathway through Scaffold Proteins

Scaffold proteins are proteins that bind to multiple partners in a signalling cascade to bring them to close proximity in a complex [113]. Many scaffold proteins have been shown to be integral for mediating signal transduction between the core components of the Hippo pathway. For example, SAV and NF2 are two scaffolds that recruit MST and LATS kinases to the plasma membrane, in turn facilitating MST-mediated LATS phosphorylation [26]. On the other hand, the WWC family of proteins (WWC1-3) are a group of cytosolic scaffold proteins involved in the Hippo signalling pathway [114]. Specifically, WWC proteins have been demonstrated to inhibit LATS1/2 autophosphorylation and bind to the PPxY motif of AMOT to prevent AMOT ubiquitination and subsequent proteasomal degradation [115]. AMOT proteins sequester YAP at the cytoskeleton to prevent YAP nuclear localization [116]. Since WBP2 interacts with the WW domain of WWC3 [72], WBP2 may similarly bind to other proteins in the WWC family through WW-mediated interactions. Furthermore, as AMOT contains PPxY motifs similar to WBP2, it is conceivable that WBP2 may compete with AMOT to regulate the Hippo/YAP signalling pathway.

Expanding the possible links between WBP2 and scaffold proteins, WBP2 has also been shown to be a binding partner of NF2, a membrane-associated scaffold [117]. Does WBP2 associate with these scaffold proteins to inhibit Hippo signalling in cancers?

## 6. Implications of the WBP2–Hippo Signalling Axis in Cancer Therapy and Precision Medicine

The rise of targeted therapeutics in the oncology market has been especially evident in recent years. This is owing to their higher efficacy and reduced side effects as compared to the classical systemic therapy. In breast cancer, the number of targeted therapeutic drugs approved by the United States of America Food and Drug Administration (USA FDA) has expounded immensely by 8 fold over the past two decades [118]. In lung cancer, a new targeted drug termed as Sotorasib was recently approved for NSCLC patients with the KRAS G12C mutation, a highly prevalent mutation that was previously deemed to be untreatable. The effectiveness of targeted therapy lies in its ability to exploit known carcinogenesis mechanisms to develop specific treatment strategy against a defined cancer subgroup. Given the dysregulation of Hippo and WBP2 signalling in cancers, targeting WBP2 may be a viable option to curb Hippo-suppressed and WBP2-driven cancer initiation and progression.

### 6.1. Targeting WBP2 Expression to Interfere with Hippo Signalling in Cancers

It is conceivable that the suppression of WBP2 is a potential onco-protective strategy to curb uncontrolled Hippo signalling. Our laboratory, along with a few others, has attempted to understand the multi-layered mechanisms regulating WBP2 expression to facilitate the rational design of WBP2 inhibitors. Here, we discuss potential avenues to inhibit WBP2 expression through transcriptional, post-transcriptional and post-translational mechanisms (Figure 8).

#### 6.1.1. Strategies to Limit WBP2 Expression at the Transcriptional and Post-Transcriptional Levels

Using a yeast one-hybrid analysis, a reporter assay, a chromatin immunoprecipitation assay and tandem mass spectrometry, our laboratory identified the upstream stimulatory factor 1 (USF1) as the transcription factor that mediates WBP2 transcription by associating with the E-box motif of the WBP2 promoter [119]. Akt phosphorylation, induced by insulin treatment, promotes USF1 activation, in turn enhancing WBP2 transcription [119]. Therefore, employing Akt inhibitors or lowering insulin levels is a plausible strategy to limit WBP2 levels.

Decoy oligonucleotides are short double-stranded deoxyribonucleic acids generated from conserved regulatory elements of target genes to recognise and block the transcription factor of interest [120]. Examples of decoy-based targeted inhibition are STAT3 antisense oligonucleotides tested for treatment of head and neck cancer and NSCLC [121,122]. Similarly, decoy oligonucleotides could be used to trap the USF1 transcription factor to indirectly target WBP2 and subvert Hippo dysregulation [123].

Besides transcriptional regulation, WBP2 expression may be controlled post-transcriptionally through miRNA inhibition. Specific miRNAs mimics could be designed to circumvent WBP2 accumulation by repressing transcription. Furthermore, MST1/2 was demonstrated to be crucial for Dicer/miRNA-mediated WBP2 downregulation [92]. However, there is currently no known drug that induces MST1/2 activation. Exploitation of cellular factors to induce MST1/2 activation and consequently WBP2 suppression are potential avenues for exploration. The concept of activating kinase for cancer treatment is not new and has been promising in recent years. For instance, triptonide elicits its anti-cancer effect in pancreatic cancer through selectively activating MEKK4 to induce the MAPKP tumour suppressive pathway [124]. In addition, the prodrugs of triptonide, minnelide have been shown to have anti-tumorigenic effects in several cancers [125,126,127,128,129] and are in various phase I and II clinical trials for acute myeloid lymphoma (NCT03347994 and NCT03760523) and pancreatic cancer (NCT03117920). It is possible that the drug hunters will eventually develop pharmacological agents against Hippo pathway kinases, such as MST1/2.

#### 6.1.2. Strategies to Limit WBP2 Expression at Post-Translational Levels

Aberrations of WBP2 not only occur at its mRNA level, but also at its protein level. An integrated proteogenomic analysis performed on breast, ovarian and colorectal cancers demonstrated only a low or moderate correlation between WBP2 mRNA and protein expression [68]. Furthermore, an analysis of WBP2 protein-mRNA correlation over a panel of 17 breast cancer cell lines revealed only a partial concordance of approximately 50% [62]. These findings suggest that a higher WBP2 mRNA expression may not necessarily translate to a higher protein expression in cancer patients. Accordingly, an evaluation of WBP2 at its protein level is likely to be more clinically relevant than assessing its mRNA level for cancer therapeutics. It is thus imperative for us to understand the post-translational mechanisms regulating WBP2 to aid the development of tailored anti-WBP2 therapeutics.

As previously mentioned, our laboratory reported that WBP2 can be subjected to proteasomal degradation upon association with ITCH E3 ubiquitin ligase [62]. However, ITCH has been found to play contrasting roles in mediating Hippo signalling and cancer promotion by degrading LATS1, a known tumour suppressor [95,96]. Hence, clarification on other post-translational modifications on WBP2 protein level may be necessary. It is possible that WBP2 is regulated post-transcriptionally by other E3 ubiquitin ligases. Initial screening through yeast two-hybrid and mass spectrometry assays performed by our group revealed E3 ubiquitin ligases such as WWP1, WWP2, HUWE1 and NEDD4L as putative WBP2 interacting partners. We concede that these notions are preliminary and need to be investigated through future studies.

Previous studies have suggested that WBP2 elicits its role as a transcriptional co-activator by migrating into the nucleus. Our laboratory has reported that ligands such as E2, EGF and Wnt can induce WBP2 phosphorylation that dictates its nuclear localization [61,62]. Indeed, the nuclear function of WBP2 is crucial for its co-activator activity with transcriptional co-factor YAP [62,73,80]. However, the majority of WBP2 actually resides in the cytoplasm, and its upstream Hippo interacting partners, WWC3 and LATS2, are also predominantly cytosolic [71,72]. Furthermore, confocal laser scanning revealed a co-localization of WBP2 and WWC3 in the cytosol of lung carcinoma cells [72]. Given that both nuclear and cytosolic WBP2 have defined molecular functions in the Hippo pathway, we postulate that directing WBP2 cellular localization is not a tenable anti-cancer avenue to modulate the Hippo pathway.

In summary, an elucidation of the key mechanisms orchestrating WBP2 expression in cancer has enabled us to design therapeutic strategies against WBP2 at transcriptional, post-transcriptional or post-translational levels. However, these therapeutic strategies are still under preclinical development, and more research must be conducted to ensure their reliability and efficiency in limiting cancer progression in the clinic.

### 6.2. Targeting WBP2 Binding with Hippo Signalling Components

The regulatory effect of WBP2 on the Hippo signalling pathway is mostly governed by its binding affinity with a myriad of regulators and effectors (Figure 6). Mapping the critical binding domains between WBP2 and its Hippo binding companions provides comprehensive information for the development of therapeutics to disrupt these protein–protein interactions, thereby decrementing WBP2-mediated molecular functions. For instance, a series of biochemical experiments in gastric cancer cells revealed that the C terminus of WBP2 is responsible for binding to the C terminal kinase domain of LATS2 [71]. In lung cancer cells, all three PY motifs of WBP2 are required for association with the WW domain of WWC3 [72]. On the other hand, the PY2 motif of WBP2 is important for binding with the WW domain of TAZ and the PY3 motif is required for interaction with YAP’s WW domain [66,130]. Collectively, these results show that the C terminus of WBP2 is imperative for its molecular role as a binding protein. A cell-permeable sequence designed to inhibit the C terminus of WBP2 can be tagged with a therapeutic protein and delivered into the tumour to disrupt the interactions between WBP2 and its binding partners.

Since the PY1 motif (PPGY) of WBP2 has a different sequence from the PY2 and PY3 motifs (PPPY), It is possible that proteins that preferentially bind to the PY1 motif of WBP2 will be discovered in future. The direct clinical implication of this is that targeting either PY motifs of WBP2 may prevent a different set of substrates from associating with WBP2, hence limiting a defined set of signalling pathways. While increasing evidence has shown that the PY motifs of WBP2 may not bind to the WW domains of its binding partners, deciphering the structural basis of PY motif-mediated interactions will facilitate the design of targeted therapeutics, such as small molecules. A recent study reported that the WW domain tandem and the PY motif tandem are essential for maximising binding affinities and target specificity, as compared to an individual domain/motif entity. This is attributed to the conformal coupling resulting from the WW domain or PY motif tandem [131]. Perhaps future biochemical studies should consider the importance of these tandem in binding affinity and specificity.

In recent years, there has been much interest in rational designs targeting the YAP/TAZ–TEAD interface. Consequently, several crystal structures of YAP/TAZ–TEAD interactions have been generated [8,9,132]. Interactions between the YAP-binding domain (YBD) of TEAD and TEAD-binding domain of (TBD) are mediated by three highly conserved interfaces, of which the third interface is the main determinant that strengthens the binding affinity [8]. Deep surface pockets revealed by the crystal structures of YAP-TEAD at the second and third interface enables the rational design of covalent inhibitor of TEAD as recently discovered by Karatas et al. (2020) [133]. Furthermore, this covalent inhibitor is able to inhibit all members of the human TEAD family, suggesting its utility across different cellular contexts [133]. In another study, verteporfin, a member of the FDA-approved compound, was found to inhibit the YAP–TEAD interaction and block tumour growth in mice [134]. Riding on the tides of designing new TEAD inhibitors, a synthetic cell penetrating peptide was recently designed and validated to block the interaction between TEAD and YAP [135]. Although TEAD has proven itself as an attractive therapeutic target due to its favourable structural and biochemical properties, the development of its inhibiting drugs are still under validation [52]. Furthermore, YAP and TAZ may associate with other binding partners such as SMAD, p73 and RUNX [136]. We posit that elucidating the structural interactions of WBP2 and YAP/TAZ may serve as a possible therapeutic direction especially in cancers with WBP2 and YAP/TAZ overexpression. While biophysical analysis has been evaluated for WBP2–YAP interactions [130], there is a pressing need for crystal structure experiments to elucidate the “druggable sites” of WBP2–YAP/TAZ interfaces.

Whether WBP2 can be truly exploited as a druggable target remains to be investigated, especially when it is an intracellular protein. It is estimated that only 10% of genes in the human genome can be targeted by drugs, while 5% are both druggable and disease-modifying [137,138]. As the oncogenic roles of WBP2 are mostly mediated through protein–protein interactions, designing small molecules that target these interactions is a plausible approach. However, an inherent challenge is the uneven distribution of affinities throughout the binding interfaces between the two interacting proteins [139,140,141]. Experiments such as alanine scanning or hot-spot analysis can be conducted to identify the exact binding regions with the highest affinity. Furthermore, three-dimensional structures of WBP2 either alone or along with its interacting partners will be useful for determining potential targetable pockets [142] although bioinformatic analysis revealed that the carboxyl terminus of WBP2 is likely to be highly flexible/unstable. Nevertheless, the challenges in targeting protein–protein interactions should not impede our attempts as every progress would contribute to future scientific breakthroughs that would resolve these roadblocks. In fact, there are already notable successes of protein–protein interaction inhibitors entering the clinic such as the PD-1/PD-L1 inhibitors [141]. We expect that novel strategies will be developed in future against protein–protein interactions that are deemed to be non-targetable, and this will benefit the development of effective WBP2 therapeutics.

### 6.3. Utilizing Anti-WBP2 Therapeutics for Combinatorial Treatment

Combinatorial therapy is the use of two or more therapies with varied mechanisms to enable effective cancer management and treatment [143]. Compared to monotherapy, combinatorial therapy is expected to reap superior anti-cancer benefits than monotherapy due to the following reasons: (1) combinatorial therapies are more likely to potentiate synergistic or addictive tumour-killing effects by targeting multiple pathways in cancer. This may prevent the likelihood of tumour cells to develop compensatory mechanisms that induce therapeutic resistance; (2) the use of FDA-approved drugs in combinatorial therapy may expedite FDA approvals for the combined treatment and reduce overall costs required for drug discovery and validation [143,144]. One successful example of combinatorial therapy in cancer is the use of trastuzumab and pertuzumab to overcome trastuzumab resistance in HER2+ breast cancer [145,146,147]. Overall, combinatorial therapy serves as a cornerstone in cancer therapeutics and is expected to provide substantial economic and clinical advantages.

The potential utilization of anti-WBP2 inhibitors as combinatorial therapy treatment is conceived from the observations that loss of WBP2 can increase sensitivity towards certain chemotherapeutic drugs such as doxorubicin, tamoxifen and trastuzumab in breast cancers [64,67,69]. Therefore, reducing WBP2 expression in clinical breast cancer patients is likely to improve patients’ response towards these anti-cancer drugs. Concomitantly, the Hippo pathway is also implicated in drug resistance mechanisms in cancers. Given the intricate link between WBP2 and Hippo signalling, targeting WBP2 can be used in combination with currently clinically available drug to minimise drug resistance and induce the Hippo pathway to curb cancer progression.

On the other hand, a number of strategies have been proposed to directly or indirectly target YAP/TAZ activity, all of which are in varying stages of development (reviewed in [52]). Furthermore, several clinically approved drugs have been reported to modulate Hippo signalling, hence providing an attractive opportunity for drug repurposing (reviewed in [148]). Some of these drugs include digitoxin and verteporfin (YAP–TEAD complex disrupters), metformin (AMOT stabiliser), dobutamine and melatonin (G-coupled protein receptor inhibitors) and getinib, erlotinib and pazopanib (tyrosine kinase receptor inhibitors) [148]. Preliminary studies from our group demonstrated that WBP2 can predict response to metformin in HER2+ breast cancer (Unpublished data). With prior knowledge that metformin may inhibit YAP nuclear translocation through the promotion AMPK-induced AMOT stabilization [149], we postulate that WBP2 can be used as a combinatorial therapy with metformin in HER2+ breast cancer [149]. This wide array of drugs constitutes an arsenal of Hippo targeting agents that could be used together with WBP2-targeting agents to achieve a combinatorial effect for cancer therapy.

## 7. Conclusions

The Hippo signalling pathway is an anti-cancer pathway with wide-ranging therapeutic implications. WBP2, first identified as a co-factor of YAP, has proven itself to be a pleotropic regulator of the Hippo pathway, amongst other pathways, by controlling the activity of Hippo core components (through LATS2 and WWC3) and its terminal effectors, YAP/TAZ. Reciprocally, WBP2 itself can be modulated by the Hippo kinase MST, signifying a positive feedback loop to turn on the switch toward carcinogenesis and tumour malignancy. Consistently, current findings on the link between WBP2 and MST/LATS/YAP have clear implications on uncontrolled cell proliferation and cancer aggressiveness. It is noted that YAP/TAZ knockout embryos suffer from premature death or polycystic kidney disease [150,151], while WBP2 knockout mice are reported to be viable with defects in hearing and other behaviours related to nervous system, metabolism and epidermal cell regeneration [56,73,152]. This suggests that WBP2 has a weaker growth phenotype than YAP/TAZ. Given the multifaceted modes of action of WBP2, we expect that further interplays between WBP2 and the Hippo pathway will be unravelled in subsequent studies. The intricate loop between WBP2 and the Hippo pathway can be potentially exploited for tailored therapeutic strategies in WBP2-driven cancers. Further characterisation of the phenotypical and mechanistic roles of WBP2 in more cancer types may expand the scope for effective WBP2 therapeutic targeting. The mechanisms on how WBP2 is regulated will also be crucial for verifying WBP2 as therapeutic target in various clinical contexts.

## Figures and Tables

**Figure 1 cells-10-03130-f001:**
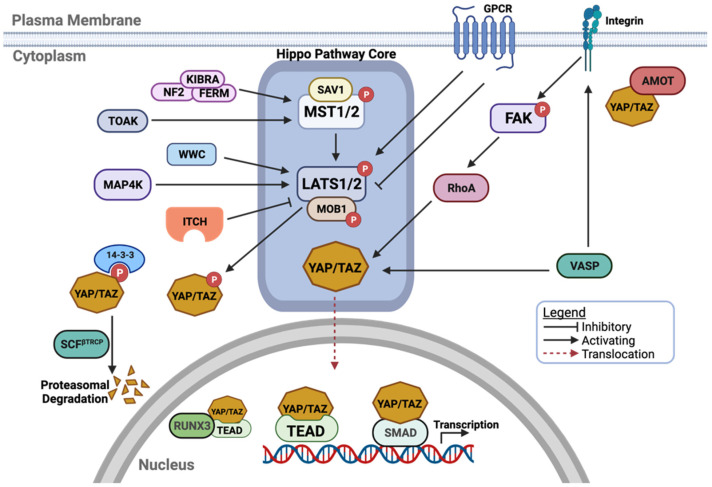
Schematic diagram of the Hippo signalling pathway. The canonical core kinase cassette is composed of the kinases MST1/2 and LATS1/2, the adaptor proteins MOB1 and SAV and the transcriptional co-factors YAP and TAZ. MST1/2 phosphorylates LATS1/2, that in turn phosphorylates and inactivates YAP. Phosphorylated YAP is sequestered by the 14-3-3 protein in the cytoplasm and degraded via proteasome facilitated by the SCF^βTRCP^ E3 ligase. Other non-canonical regulators have been discovered to modulate the Hippo pathway core. The KIBRA–NF2–FERM complex and TOAK has been identified to play an activating role on MST1/2. WWC is an adaptor protein that binds and activates LATS while MAP4K may phosphorylate LATS directly. ITCH is a E3 ligase that promotes ubiquitination and proteasomal degradation of the LATS. AMOT directly binds to YAP/TAZ to promote YAP/TAZ cytoplasmic sequestration and phosphorylation. Integrin signalling is another non-canonical pathway that activates YAP through FAK/RhoA and VASP. Additionally, VASP can stabilise YAP/TAZ by inducing its dephosphorylation. YAP/TAZ associates with transcription factors such as TEAD and SMAD in the nucleus where they initiate gene transcription. Activity of the YAP/TAZ–TEAD complex in the nucleus may be inhibited by its binding with RUNX3. Created with BioRender.com.

**Figure 2 cells-10-03130-f002:**
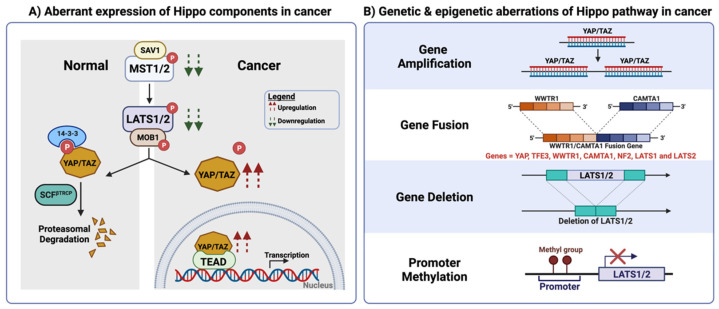
Aberration of the core Hippo pathway components in cancer. (**A**) Aberrant expression of Hippo components in cancer. Tumour suppressor such as MST/LATS were found to be downregulated at the mRNA and protein level in various cancers, while oncogenic YAP/TAZ were found to be upregulated with tumour progression. (**B**) Genetic aberrations of the Hippo pathway in cancer. Examples of genetic aberrations: gene amplification, gene fusion, gene deletion and promoter methylation. Created with BioRender.com.

**Figure 3 cells-10-03130-f003:**
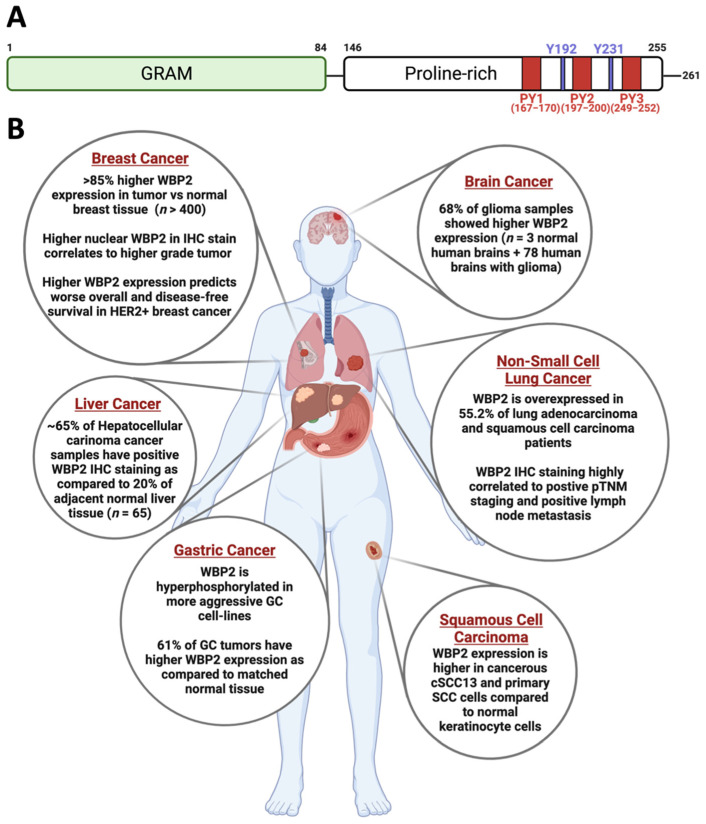
Structure and implications of WBP2 in various cancer types. (**A**) Schematic diagram of the WBP2 domain structure. WBP2 is comprised of the GRAM domain at the N terminus (1–84 amino acid (a.a)) and the proline-rich region at the C terminus (146–261 a.a). There are three PPxY motifs namely PY1 (167–170 a.a), PY2 (197–200 a.a) and PY3 (249–252 a.a) that have been identified as being involved in protein–protein interactions. Two phospho-tyrosine sites at Y192 and Y231 are found in the proline-rich region of WBP2. Phosphorylation of WBP2 is important for its nuclear translocation. (**B**) Implications of WBP2 in various cancer types based on the current literature. Created with BioRender.com.

**Figure 4 cells-10-03130-f004:**
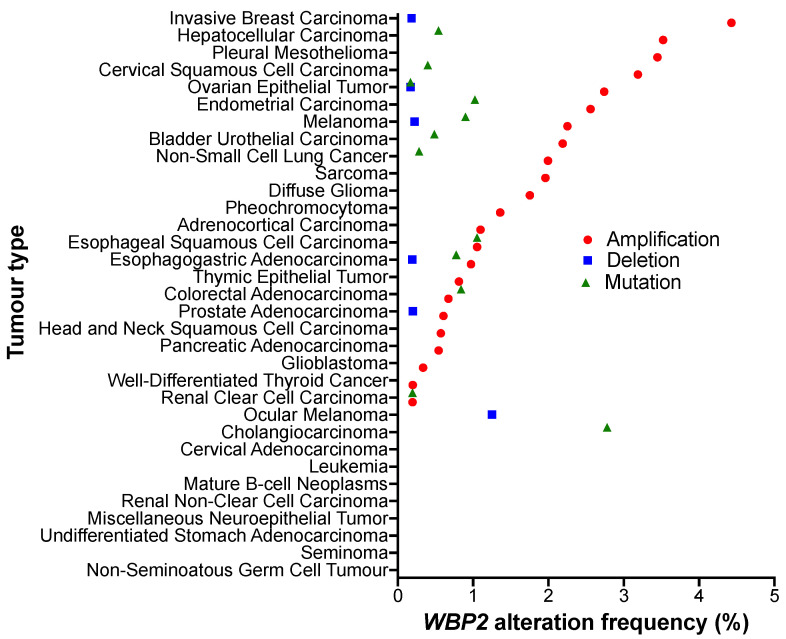
WBP2 copy number alteration frequency in 32 cancer types from the TCGA Pan-cancer Atlas. The x-axis shows the frequency of WBP2 alteration and the y-axis shows the tumour types found in the TCGA Pan-Cancer Atlas [75]. Each dot represents the frequency of the indicated types of WBP2 alteration; a red circled dot represents amplification, a blue squared dot represents deletion and a green triangle dot represents mutation. Data were downloaded from cBioportal [76,77] and created using GraphPad Prism.

**Figure 5 cells-10-03130-f005:**
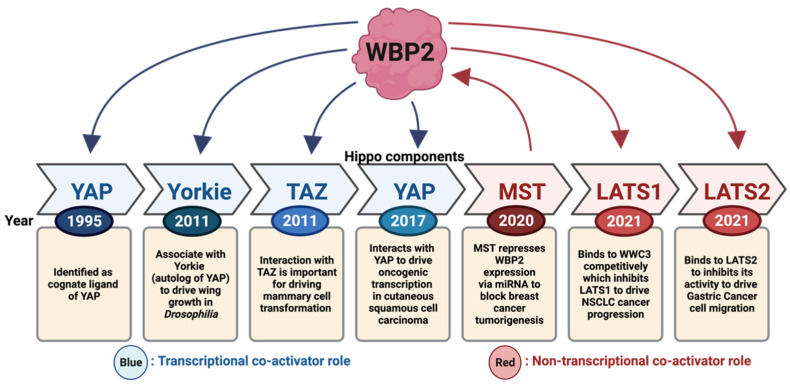
Schematic timeline of WBP2 implication in the Hippo pathway. WBP2 was first discovered to interact with YAP in 1995. Later in 2011, it was shown to promote growth in Drosophila by associating with Yorkie. In the same year, its oncogenic role affecting the Hippo pathway was first implicated in breast cancer through its association with TAZ. The oncogenic role of WBP2 in directing Hippo dysregulation was later extended to other cancer types including cutaneous squamous cell carcinoma (cSCC), non-small cell lung carcinoma (NSCLC) and gastric cancer. The Hippo pathway regulating WBP2 levels was identified to be through the Dicer–miRNA axis by MST. The text coloured in blue describes the transcriptional co-activator role of WBP2 in the Hippo pathway through binding to YAP and TAZ. The text coloured in red describes a non-transcriptional co-activator mechanism of WBP2 in the Hippo pathway. The arrow indicates the direction of regulation (i.e., WBP2 on Hippo/Hippo on WBP2). Created with BioRender.com.

**Figure 6 cells-10-03130-f006:**
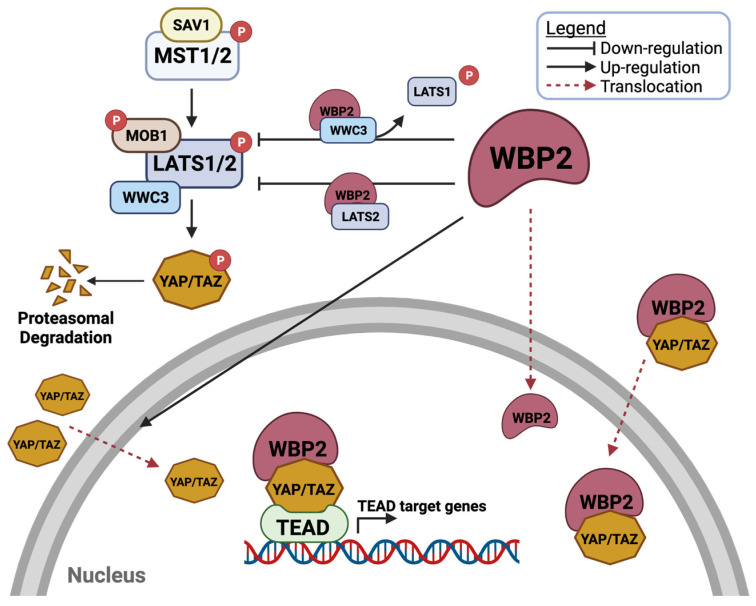
Regulation of the Hippo signalling pathway by WBP2. WBP2 interacts with LATS2 to inhibit the LATS2-mediated phosphorylation of YAP. WBP2 binds competitively with WWC3 to displace LATS1, hence limiting LATS1 activation. WBP2 translocates into the nucleus and interacts with YAP/TAZ to activate target gene transcription. Created with BioRender.com.

**Figure 7 cells-10-03130-f007:**
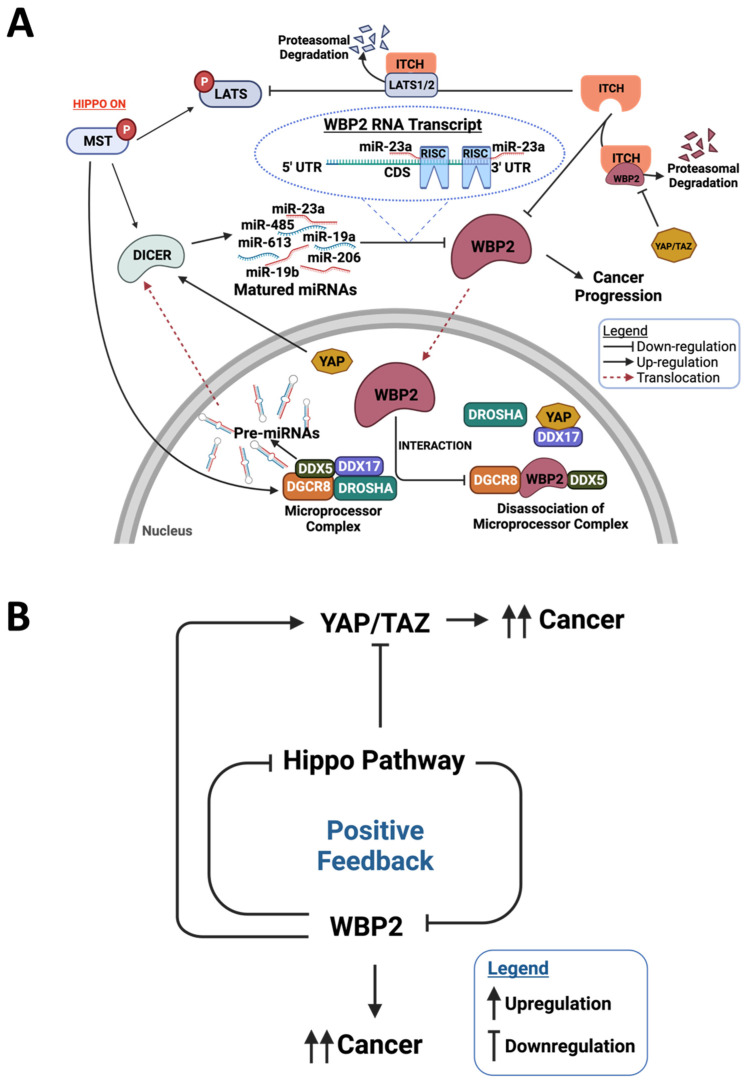
Regulation of WBP2 by the Hippo signalling pathway. (**A**) Overview of the modes of action of how WBP2 can be regulated by Hippo pathway in cancer. In the nucleus, WBP2 negatively regulates miRNA biogenesis through its interaction with MPC components, hence disrupting the MPC complex and its miRNA processing capacity. On the other hand, the nuclear accumulation of YAP and the activation of MST maintains the high expression of Dicer. Pre-miRNAs are processed by Dicer in the cytosol to mature miRNAs in turns target WBP2. Some miRNAs known to target WBP2 include miR-19a/b, miR-23a, miR-27a, miR-206, miR-485 and miR-613. WBP2 level is also regulated by proteasomal degradation where the interaction of WBP2 with ITCH via the WW-PPxY domain facilitates the ubiquitin-mediated proteasomal degradation of WBP2. ITCH E3 ubiquitin ligase promotes the ubiquitin-mediated proteasomal degradation of LATS. (**B**) A positive feedback loop in the regulation of WBP2 and the Hippo signalling pathway drives cancer progression. Created with BioRender.com.

**Figure 8 cells-10-03130-f008:**
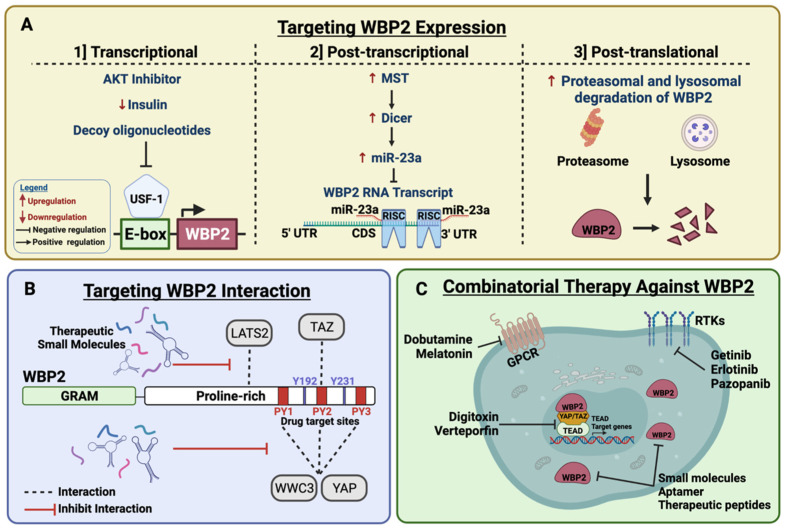
Summary of WBP2 translational implications in Hippo-dysregulated cancers. (**A**) Targeting WBP2 expression could be carried out at the transcriptional, post-transcriptional and post-translation levels. (**B**) Targeting WBP2 interacting partners could be facilitated by the design of small molecules that target the PPxY motifs in the C terminus of WBP2. This is expected to block the interaction of WBP2 with its interacting partners in the Hippo pathway to subvert WBP2-driven cancer progression. (**C**) Combinatorial therapy involving potential WBP2 inhibitors and other therapeutics to improve treatment efficacy and reduce possible side effects. Created with BioRender.com.

## Data Availability

Not applicable.

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
