# Peer review of "Reciprocal Regulation of Hippo and WBP2 Signalling—Implications in Cancer Therapy"

_cells, 2021, doi:10.3390/cells10113130_

Round 1

Reviewer 1 Report

The authors summarized the interplay between Hippo pathway and their biological mechanism. They also proposed WBP2-Hippo inteaction as a therapeutic target for cancer treatment.

WBP2 was indentified as a cognate ligand of YAP in 1995. Although WBP2 are recently discovered to modulate the Hippo signaling pathway, only a few groups including Lim YP's reported the interplay between WBP2 and Hippo pathway.

Also, mouse genetic data of WBP2 is somewhat different from those of Hipp pahway. 
YAP and Lats knockout embryos die during early embryogenesis and TAZ knockout display polycystic kidney or premature death. However, WBP2 knockout mouse is viable and show defect in hearing and behavior. 
Therefore, furhter studies and evidence of the interaplay are required to propose WBP2 as a reliable, real regulator of Hippo pathway.

Author Response

We thank the reviewer for the valuable feedback Although WBP2 was discovered to be a cognate ligand of YAP in 1995, and while WBP2’s binding with YAP/TAZ has been reported by us and others subsequently, the authors concede that the association of WBP2 with other core Hippo components, namely LATS1/2 and MST1/2, are more recent findings. Notwithstanding that WBP2 has been implicated in the Hippo pathway in at least three different cancer types, including breast, gastric and lung cancers by independent groups , further investigations should clarify the role of WBP2 in the Hippo pathway. We have added a relevant discussion on this in the manuscript (pg 10 line 348-356 and pg 20 line 738-742).

Reviewer 2 Report

This is an up-to-date review on one of the facets of the signalling machinery of the Hippo pathway, with particular reference to the interplays between YAP/TAZ and WBP2 and some of their molecular regulators. The authors have chosen to set up the manuscript layout based upon partition in sequential subheadings. Although this may in some circumstances be useful for readers, I fear that in this case it is simply confusing and in some cases not proper.

Starting from the initial sections of the manuscript, an alternative layout may be to initiate Section 2 related to the role of WBP2 in cancer with the description of its expression pattern in the different cancer types and the implication of such expression without subheadings. Thus, a major heading should be number 2, with a more specific reference to "WBP2 expression and impact in cancer". By costrast, separation in subheadings 2.1, 2.1.1 and 2.2.2, 2.3.1 etc. is superfluous and could be omitted. This section should be written as a continous text with no separation into subheadings.

Subheading 3 would then become another major heading and the more detailed subheadings 3.1.1, 3.1.2. etc. could similarly be omitted. Subheading 3.2.1. would then become a major heading, as number 4 and the 3.2.2. subheading converted into heading number 5. The same applies to the subsequent heading/subheading separations.

The Sections 5.1.1 and 5.1.2. should be reformulated and entirely revised since they are confusing, difficult to follow and do not provide a clear message.

Figure 7 is informative, but the schemes "B" and "C", which are rather central to the discussions made in tha part of the text, should be significantly improved, both in their content and the layout, such as to better assist the interpretation of the outlined notions. Preferably, a separate figure should be introduced to better illustrate and more clearly highlight the putative modes through which WBP2 may be targeted therapeutically.

Finally, Section 5.3 should be considerably pruned.     

Author Response

We thank the reviewer for the feedback. We have considered the reviewers’ comments and imposed a few edits to improve our manuscript:

  • For Section 2.2, we have edited the heading to “WBP2 expression and impact in cancers” (pg 5 line 161). We also removed the subheadings of 2.2.1 and 2.2.1 and presented Section 2.2 as a continuous text
  • We have re-formulated the original Section 3.1 (Regulation of the Hippo pathway by WBP2) to major Section 3 and the original Section 3.2 (Regulation of WBP2 by the Hippo pathway) to major Section 4. The rest of the subsections has been renumbered accordingly. Although the reviewer suggested the original subheading 3.2.2 (Now Section 4.2) to be relabelled as a major Section 5 on its own, we feel that this subsection is an elaboration of major Section 4, and hence we decide to continue to group it under Section 4.
  • Section 6.1.1 and Section 6.1.2 (previously Section 5.1.1 and Section 5.1.2) describes some of the therapeutic strategies to reduce WBP2 expression transcriptionally/post-transcriptionally and post-translationally. We have renamed the headings of subsection 6.1.1 and 6.1.2 to reduce confusion. We have also added a short paragraph at the end of subsection 6.1 (pg 17-18 line 620-624) to provide better clarity of the subsection.
  • Figure 8 (previously figure 7), is meant to be an overview of Section 6. We have edited Figure 8B to improve the clarity of the figure.
  • We have trimmed Section 6.3 (previously 5.3) considerably

Reviewer 3 Report

In the review entitled "Reciprocal Regulation of Hippo and WBP2 Signalling - Implications in Cancer Therapy" by Lim YX et al, authors make an extensive revision regarding the cross-talks of the hippo signalling and the oncogenic  WBP2. They thoroughly address all the connections between this co-factor and the different members of the hippo pathway reciprocally, since the interplay between them seems to be taking place in both ways. Also, they explain the potential of exploiting this interconnection in order to tackle cancer, opening new unaddressed questions within the field. The text is very well written and organized, accompanied with several figures that support and clarify the main ideas. 

As a minor suggestion, I would like to ask the authors to include a bit more regarding the interaction of YAP with p73 and its role as tumour suppressor, and if there is any evidence or suggestion indicating that WBP2 could regulate or interfere with this function of YAP.

Author Response

We thank the reviewer for his time and positive feedback for our manuscript. We have considered the reviewer’s suggestion on YAP-p73 interaction and discussed it in our manuscript (pg 9 line 317-324)  

Round 2

Reviewer 2 Report

Re-organization and improvement of the manuscript layout has been carried out properly and the review is now therefore in its publishable format.